# Time-Course Transcriptome Profiling Reveals Differential Resistance Responses of Tomato to a Phytotoxic Effector of the Pathogenic Oomycete *Phytophthora cactorum*

**DOI:** 10.3390/plants12040883

**Published:** 2023-02-15

**Authors:** Xue Zhou, Ke Wen, Shen-Xin Huang, Yi Lu, Yang Liu, Jing-Hao Jin, Shiv D. Kale, Xiao-Ren Chen

**Affiliations:** 1College of Plant Protection, Yangzhou University, 48 Eastern Wenhui Road, Yangzhou 225009, China; 2Joint International Research Laboratory of Agriculture and Agri-Product Safety of Ministry of Education of China, Yangzhou University, 48 Eastern Wenhui Road, Yangzhou 225009, China; 3Fralin Life Science Institute, Virginia Tech, Blacksburg, VA 24060, USA

**Keywords:** tomato, transcriptome, *Phytophthora cactorum*, small cysteine-rich protein, phytotoxicity, defense response

## Abstract

Blight caused by *Phytophthora* pathogens has a devastating impact on crop production. *Phytophthora* species secrete an array of effectors, such as *Phytophthora cactorum*-*Fragaria* (PcF)/small cysteine-rich (SCR) phytotoxic proteins, to facilitate their infections. Understanding host responses to such proteins is essential to developing next-generation crop resistance. Our previous work identified a small, 8.1 kDa protein, SCR96, as an important virulence factor in *Phytophthora cactorum*. Host responses to SCR96 remain obscure. Here, we analyzed the effect of SCR96 on the resistance of tomato treated with this recombinant protein purified from yeast cells. A temporal transcriptome analysis of tomato leaves infiltrated with 500 nM SCR96 for 0, 3, 6, and 12 h was performed using RNA-Seq. In total, 36,779 genes, including 2704 novel ones, were detected, of which 32,640 (88.7%) were annotated. As a whole, 5929 non-redundant genes were found to be significantly co-upregulated in SCR96-treated leaves (3, 6, 12 h) compared to the control (0 h). The combination of annotation, enrichment, and clustering analyses showed significant changes in expression beginning at 3 h after treatment in genes associated with defense and metabolism pathways, as well as temporal transcriptional accumulation patterns. Noticeably, the expression levels of resistance-related genes encoding receptor-like kinases/proteins, resistance proteins, mitogen-activated protein kinases (MAPKs), transcription factors, pathogenesis-related proteins, and transport proteins were significantly affected by SCR96. Quantitative reverse transcription PCR (qRT-PCR) validated the transcript changes in the 12 selected genes. Our analysis provides novel information that can help delineate the molecular mechanism and components of plant responses to effectors, which will be useful for the development of resistant crops.

## 1. Introduction

Oomycota is a phylum of eukaryotic microorganisms in the kingdom Stramenopila that are phylogenetically close to photosynthetic algae [1]. The phylum consists of many notorious phytopathogens, including approximately 100 species within the genus *Phytophthora.* A large number of *Phytophthora* species are well-known agents of broad and specific crop and ecosystem destruction. For example, *Phytophthora infestans* is the causal agent of potato and tomato late blight and was responsible for the Irish potato famine in the mid-19th century. *Phytophthora cactorum* can infect an extremely wide range of plant species, including horticultural crops and trees [2]. Pathogenesis is thought to be facilitated in part by secreted proteins, known commonly as effectors, which can function through a variety of mechanisms. In many cases, effectors can be directly or indirectly recognized, resulting in the mounting of a defense response by the host plant [3]. Several virulence factors—notably, SCR96—have been shown to possess phytotoxic properties [4,5,6]. A temporal understanding of host transcription changes in response to such virulence factors can facilitate insights into which pathways and gene families are being activated and silenced.

Several groups of apoplastic effectors from plant pathogenic oomycetes have been identified, including elicitins, the Nep1-like (NLP) family of proteins, *P. cactorum-Fragaria* (PcF)/small cysteine-rich (SCR) proteins, proteinaceous enzyme inhibitors, the GP42 (PEP13) transglutaminase, and cellulose-binding elicitor lectin (CBEL) glycoproteins [3]. Some of these effectors are conserved, such as NLPs and elicitins. These effectors can be recognized by receptor-like proteins (RLPs)—i.e., RLP23 and ELR, respectively—in a manner similar to pathogen-associated molecular patterns (PAMPs) [7,8,9]. Little is known about the functions and roles of most apoplastic effectors, including PcF/SCRs. This family of effectors—often identified with the abbreviation “SCR” for “small cysteine-rich”, followed by their predicted amino acid length—were initially regarded as oomycete “orphan” avirulence (Avr) proteins for which the corresponding resistance gene had not been identified [10]. Currently, PcF/SCRs are thought to function as extracellular toxins based on the fact that the members identified so far can cause plant leaf necrosis, such as the first member PcF and SCR96 from *P. cactorum* [4,11,12,13]. Mature PcF is an acidic protein with 52 amino acids and contains six cysteine residues bridged intramolecularly. It causes leaf necrosis and defense-related responses in strawberry and tomato [11,14]. A number of SCR proteins with the PcF domain (Pfam PF09461), including SCR74, have been annotated in different oomycete species [10,15]. The gene *scr74* belongs to a highly polymorphic gene family with signatures of positive selection in *P. infestans* [15]. In contrast, in *P. cactorum*, no sequence polymorphism has been observed for SCR96, which lacks the PcF domain. Expression of *scr96* is significantly upregulated during the early infection stages in host plants. SCR96 induces cell death (necrosis) responses in the Solanaceae family, including *Nicotiana benthamiana* and tomato. Gene silencing analysis showed that SCR96 is important for the pathogenicity of *P. cactorum* in plants [4]. Recently, we showed that one PcF/SCR effector in *P. capsici*—i.e., SCR82—functions as both plant defense elicitor and virulence factor [16]. So far, their targets or receptors in plants have not been determined, although SCR74 receptor was recently mapped to a 43 kbp G-type lectin receptor kinase (*G-LecRK*) locus in wild potato [17]. The mechanism behind such phytotoxic effectors remains obscure.

Programmed cell death in response to pathogen invasion has long been considered a resistance phenotype. However, in certain pathosystems, cell death is often uncoupled from resistance [18]. Recently, Wang et al. (2021) showed that sORF-encoded polypeptide SEP1 can induce cell death in *N. benthamiana* and is a virulence factor of *Phytophthora* pathogens [19], which is similar to the case of *P. capsici* SCR82 [16]. Some necrotrophic pathogens, such as *Cochliobolus victoriae*, secrete toxins to hijack the plant cell death mechanism in order to kill plant cells and feed on the cell debris [20]. Our previous work showed that silencing of *scr96* in *P. cactorum* significantly affected the pathogen virulence in plants, suggesting an important virulence role for SCR96 during infection [4]. Our working model suggests that the cell death (necrosis) induced by SCR96 may be cytolytic and that the cytotoxic effect of SCR96 may not be a part of a typical PAMP-triggered immunity (PTI) response.

We conducted a temporal transcriptomic analysis (RNA-Seq) of tomato leaves treated with SCR96 to explore unknown host responses. To perform protein treatment, SCR96 was first expressed and purified from *Pichia pastoris* PichiaPink Strain 4. To investigate the temporal expression patterns of genes involved in the plant defense, tomato leaves were treated with a low concentration of 500 nM SCR96 for 0, 3, 6, and 12 h (prior to the visual development of necrosis). Expression changes after treatment in tomato genes associated with defense mechanisms and metabolism pathways, such as plant–pathogen interaction, mitogen-activated protein kinase (MAPK) signaling, and transport pathways, were observed, which demonstrated temporal transcriptional accumulation patterns. Twelve of these genes were randomly selected and validated using quantitative reverse transcription PCR (qRT-PCR) over the time course. Potential genes involved in physiological reactions downstream of SCR96-induced responses were identified via transcriptome comparison, including those encoding receptor-like kinases (RLKs)/receptor-like proteins (RLPs), G-LecRK, resistance (R) proteins, MAPKs, transcription factors (TFs), pathogenesis-related (PR) proteins, and transport proteins. This study is the first analysis of plant responses to phytotoxic PcF/SCR effectors using RNA-Seq and may provide insights into the molecular mechanism by which these effectors impact plant–*Phytophthora* interactions. The knowledge obtained from the study could potentially be useful in the development of *Phytophthora* blight-resistance crops.

## 2. Results

### 2.1. The Recombinant SCR96 Protein Secreted by Yeast Cells Can Cause Tomato Leaf Necrosis

The small recombinant SCR96 protein was generated using *P. pastoris* via the Invitrogen PichiaPink expression system. A pilot expression experiment using nine positive colonies was performed to determine the optimal conditions for protein expression (Appendix A). One recombinant clone, P3, consistently produced protein yields higher than others. Peak production was observed at 72 h post-methanol induction. This recombinant clone and time point were chosen for large-scale expression; affinity purification yielded the recombinant protein with high purity for use in the following analyses (Appendix A).

Our previous work showed that transient expression of SCR96 by agroinfiltration and recombinant protein purified from mammalian cells could cause necrosis in tomato leaves [4,5]. To check protein activity here, the recombinant SCR96 protein (rSCR96) was infiltrated into tomato leaves. We observed that rSCR96 proteins with C-terminal His9 tags were also able to cause necrosis in tomato leaves at 12 h in a dose-dependent manner (Figure 1). For the following analyses, we chose 500 nM as the treatment concentration for rSCR96 due to it consistently producing leaf necrosis visible to the naked eye.

### 2.2. RNA Sequencing and Mapping to the Reference Genome

To explore how tomato responds to the phytotoxic effector SCR96 at the transcriptional level, tomato leaves were treated with rSCR96 and subjected to transcriptome analysis at four time-points (0, 3, 6, and 12 h post-treatment (hpt)). All time-point treatments were analyzed in three independent biological replicates (12 samples in total). Approximately 25 million paired end reads were generated per sample library. The GC content of the post-trimmed reads varied from 42% to 44% in the libraries, and Q30% was greater than 93%, indicating that the reads were of high quality (Appendix A).

The number of quality-assessed reads uniquely mapped to the tomato reference genome ITAG4.0 assembly ranged between 96.9% and 98.5% of total reads (Appendix A). Pearson’s correlation coefficients, used to assess biological reproducibility, were approximately 1, indicating that the correlation between the biological replicates of the different samples was high (Appendix A). After de novo assembly using the software StringTie [21], 36,779 unigenes (File S1) were obtained, including 2704 novel genes (Appendix A). To obtain functional information for the unigenes, gene annotation was performed using seven public databases: Cluster of Orthologous Groups of Proteins (COG), Gene Ontology (GO), Eukaryotic Orthologous Groups (KOG), the Kyoto Encyclopedia of Genes and Genomes (KEGG), Pfam, Swiss-Prot, and Non-Redundant Protein Sequence Database (NR). A total of 32,640 (88.7%) unigenes were annotated in at least one database (Appendix A).

### 2.3. Identification of Differentially Expressed Genes

Differential expression analysis for all possible pairs of time points was performed to assess the temporal alterations in the gene expression of tomato affected by SCR96 over the time course (Figure 2). SCR96 challenge caused an immediate increase in the number of differentially expressed tomato genes (DEGs) at 3 hpt in comparison to control (Figure 2a). It is important to note the larger number of DEGs at the early treatment time-point (3 hpt) compared to the later treatment time-points (6 and 12 hpt). It is also essential to note that gene expression changes lasted for the whole time course, suggesting continuous modulation of gene expression. We also observed slightly more downregulated DEGs than upregulated DEGs in each of the three comparison-to-control datasets (H3 vs. H0, H6 vs. H0, H12 vs. H0) (Figure 2a).

Multiple comparisons of DEG sets between each time point were conducted and visualized. The plot showed that many DEGs were comparison set-specific, including 1783 genes in set H3 vs. H0, 441 genes in H6 vs. H0, and 453 genes in H12 vs. H0 (Figure 2b). Upregulated genes in comparisons of H12 vs. H0, H6 vs. H0, and H3 vs. H0 were combined as DEGs co-upregulated by SCR96 treatment. As a whole, a total of 5929 non-redundant genes were found to be significantly co-upregulated in common in SCR96-treated leaves compared to untreated control leaves across all three time points (3, 6, 12 hpt) (Appendix A).

Hierarchical clustering further demonstrated that DEGs from the comparison datasets (H3 vs. H0, H6 vs. H0, H12 vs. H0) were clustered into groups based on expression changes in similar directions, although the expression levels were different across time points. The expression profiles of the DEGs at the three treatment time-points were highly correlated with each other compared to those in the control (Figure 3). The data together indicated that there was a varying temporal expression pattern for tomato genes after challenge with SCR96 that lasted over the time course.

### 2.4. qRT-PCR Analysis of the Gene Expression Profiles

We performed a qRT-PCR assay with 12 arbitrarily selected DEGs to assess the reliability of the RNA-Seq study (Appendix A). These genes consisted of nine and three genes that were upregulated and downregulated, respectively, by SCR96, including four related to biotic stress, four related to metabolism, two related to photosystems, and two that were uncharacterized (Appendix A). Consistent with the transcriptomic data, nine genes showed increased expression levels when the treatment occurred—in particular, the biotic stress-related genes (*Solyc05g014590.3, Solyc11g072930.3, Solyc08g080660.1*, and *Solyc03g020050.3*)—while the expression levels of three genes (*Solyc09g010530.3, Solyc02g071000.1*, and *Solyc02g070970.1*) were suppressed by SCR96. The expression patterns of each gene checked with the qRT-PCR were similar to the RNA-Seq data, confirming the accuracy and reproducibility of our study results (Figure 4). It was noted that, among the nine upregulated genes, the one (*Solyc08g080660.1*) encoding the PR-5 precursor was continuously responsive to the treatment over the time course; the expression of another gene (*Solyc11g072930.3*) encoding a leucine-rich repeat RLK peaked at 3 hpt but decreased gradually afterwards.

### 2.5. Temporal Clustering Analysis to Predict Putative Co-Regulated Genes

The differential analysis conducted with the RNA-Seq data separated transcripts based on statistically significant differential accumulation at specific time points but did not consider the accumulation pattern across different time points. Genes with similar expression patterns may share a similar regulatory mechanism, which could possibly place the transcripts in the same transcription factor regulatory network. To identify transcripts with similar accumulation profiles during elicitation, a co-expression analysis using the Clust tool [22] was performed for all SCR96-responsive transcripts.

According to the Clust clustering algorithm, the SCR96-responsive transcripts could be assigned to a single model profile. The Clust analysis yielded 14 major group clusters (Figure 5; Appendix A). In general, the response to SCR96 was not restricted to transcripts within a certain type of accumulation pattern but widely covered transcripts displaying different types of temporal profiles. SCR96-dependent transcripts showed higher representation in some clusters, such as clusters 0 and 7. The former (5174 genes) showed early induced and continuous responses, while the latter (3938 genes) was an example of the repressed response of tomato genes to SCR96 challenge. Other clusters containing fewer transcripts were further grouped into the below types: (i) early repressed and late induced (clusters 8, 9, and 10); (ii) early induced and intermediately repressed (clusters 1, 2, 3, and 4); (iii) nearly constantly repressed (clusters 5 and 6); (iv) early constant and late induced (clusters 11 and 12); and (v) early induced and sustained (cluster 13). Such clusters displayed differences in gene expression patterns, although they were roughly classified into the same types. The temporal analysis via clustering suggested multiple types of tomato transcriptional response to the phytotoxic protein SCR96.

### 2.6. GO and KEGG Analyses Revealed Genes Involved in Plant Defense Responses and Metabolism

In contrast to the control, many genes were found to be responsive to SCR96; in particular, those identified in the three comparison sets (H3 vs. H0, H6 vs. H0, H12 vs. H0). Enriched GO terms from the biological processes categories were found to be held in common in these sets, including response to biotic stimulus (GO:0009607), cellular process regulating host cell cycle in response to virus (GO:0060154), and immune system process (GO:0002376). For the enrichment of GO cellular components, the results confirmed activities denoted by common enriched GO terms occurring in the extracellular space, membrane, and other cellular compartments, such as the extracellular region (GO:0005576), membrane (GO:0016020), and organelle (GO:0043226) terms. In the molecular function category, genes associated with the binding (GO:0000989, GO:0005102, GO:0030674, GO:0001664, and GO:0003676), catalytic activity (GO:0003824), and transporter activity (GO:0005215) terms were enriched (Table 1). Moreover, the KEGG pathway enrichment analysis showed that plant–pathogen interaction (map04626), protein processing in the endoplasmic reticulum (map04141), MAPK signaling pathway—plant (map04016), and endocytosis (map04144) were the most commonly overrepresented in the DEGs from all comparison groups (Appendix A). However, other KEGG pathways were highly enriched in specific comparison groups, such as spliceosome (map03040), RNA transport (map03013), and several others for H3 vs. H0 (Appendix A). Taken together, the annotation of DEGs revealed the transcriptional changes in many defense-related and metabolic genes in response to SCR96 treatment.

### 2.7. Genes Involved in Pathogen Detection and Signal Transduction

Many of the host genes showed enriched processes related to plant–pathogen interactions. The annotation of all DEGs demonstrated that there were 224 genes encoding RLKs and 25 RLPs (Appendix A). Generally, these plant surface receptors can recognize apoplastic effectors, PAMPs, microbe-associated molecular patterns (MAMPs), and damage-associated molecular patterns (DAMPs), initiating a general signal transduction cascade of defense responses [23]. Further investigation showed that about half of the RLK genes were significantly induced, while the remainder were reduced by SCR96. For example, *Solyc11g072930.3*, encoding one RLK, showed rapid induction after SCR96 treatment that persisted over the time course (Figure 4). Most of the *RLP* genes were significantly upregulated by SCR96, with varied expression by the end of the time course (Appendix A). Calcium ion flux at the plasma membrane, which is part of PTI, has been found to be involved in plant–microbe interactions [24]. It was found here that, among the *RLK* genes, five related to calcium signaling were significantly regulated by SCR96, with two being downregulated (*Solyc07g007020.3* and *Solyc09g014740.3*) and three upregulated (*Solyc02g089290.3, Solyc09g014720.3*, and *Solyc11g072140.3*) (Appendix A). In addition to PTI, the intracellular recognition reaction is triggered by the perception of effectors by plant nucleotide-binding/leucine-rich repeat (NLR) receptors (i.e., disease resistance proteins, R proteins), leading to effector-triggered immunity (ETI) and, therefore, greater robustness in relation to pathogen disturbance. The induction of 43 genes encoding NLRs was detected upon SCR96 treatment (Appendix A), suggesting that ETI may also be triggered by SCR96.

Upon pathogen recognition, different host signaling cascades were initiated to promote local and systemic defense responses, including MAPK cascades [25]. To identify important MAPK signaling cascades responsive to SCR96, we focused on expression changes in all MAPKKK, MEK, and MPK genes (collectively referred to as MAPK genes). In total, 30, 19, and 17 MAPK genes showed significant changes in transcript abundance in response to SCR96 at 3, 6, and 12 hpt, respectively (Appendix A). We found a very strong and immediate induction of MAPK genes (23 upregulated vs. 7 downregulated) by SCR96 at 3 hpt (Appendix A). At the level of individual genes, we observed an overlap consisting of 12 genes between the H3 vs. H0, H6 vs. H0, and H12 vs. H0 comparison groups. Of these 12 genes, 8 (*Solyc04g007710.3, Solyc12g040680.2, Solyc02g084870.3, Solyc02g090980.1, Solyc08g081490.4, Solyc12g088940.3, Solyc01g094960.3*, and *Solyc11g072630.2*) showed significant upregulation after SCR96 treatment, irrespective of time points.

### 2.8. G-LecRK Gene Expression Changes

The immune receptors of PcF/SCR effectors in plants have not been identified. Recently, the SCR74 response was mapped to a *G-LecRK* locus in the wild potato *Solanum microdontum* subsp. *gigantophyllum* [17,26]. Since SCR96 is regarded as a PcF/SCR effector, this prompted us to test whether *G-LecRK* genes respond to SCR96 treatment. This study revealed that a total of 44 *G-LecRK* genes at 3 hpt, 39 *G-LecRK* genes at 6 hpt, and 33 *G-LecRK* genes at 12 hpt were significantly upregulated compared with 0 hpt (Figure 6a; Appendix A). These accounted for 37 unique DEGs across the three comparison sets (Figure 6b; Appendix A). The expression heat map of all 37 DEGs demonstrated that the majority of DEGs were induced in response to SCR96 (Figure 6c). These DEGs were clustered into groups based on expression changes, but the expression levels varied across the four time points. Five genes (*Solyc02g079570.4, Solyc03g005130.3, Solyc05g008310.4, Solyc11g013880.1*, and *Solyc12g006840.2*) demonstrated the most significant upregulation upon SCR96 treatment, especially at the earliest time point (3 hpt) (Figure 6c).

### 2.9. Activation of TF Genes

TFs are pivotal regulatory proteins in signal transduction networks activated in plants in response to various stresses. We analyzed the time-dependent regulation of genes encoding TFs in tomato after SCR96 treatment. Based on sequence similarity, the genes involved in transcriptional regulation were classified by family. In total, 76 TF families consisting of 2410 genes were identified in this study. Major TF families consisting of more than 100 members included the myeloblastosis (MYB)-related family; APETALA2/ethylene-responsive factor (AP2/ERF) family; basic helix loop helix (bHLH) family; cysteine-2/histidine-2 (C2H2) family; MCM1, AGAMOUS, DEFICIENS, and SRF (MADS) family; and homeobox (HB) family (Appendix A). After SCR96 challenge, compared to the control (0 hpt), a total of 769, 650, and 613 TF genes were differentially expressed at 3, 6, and 12 hpt, respectively (Appendix A). For example, one bHLH family member, *Solyc05g014590.3*, showed a rapid response to SCR96 challenge, which lasted for the entire time course (Figure 4). Based on the set comparisons, the six families were identified as significantly participating in transcriptional regulation after treatment, and most TF genes were differentially expressed at 6 hpt (Figure 7). The strong induction of up to 40 AP2/ERF genes at 6 hpt was remarkable, as was the expression repression of 36 AP2/ERF genes. In general, fewer TF genes were differentially regulated at 12 hpt compared to the other two time points.

### 2.10. Genes Encoding PR Proteins

PR proteins have been reported to possess antimicrobial activities that occur via damaging action affecting the cellular structures of parasites [27]. In total, 28, 30, and 26 PR genes were detected in response to SCR96 at 3, 6, and 12 hpt, respectively (Appendix A; Appendix A). Among these, 19 genes were commonly regulated in samples at these three time points. A total of 16, 24, and 18 genes were upregulated in the 3, 6, and 12 hpt samples, respectively. qRT-PCR analysis revealed that the expression of *Solyc08g080660.1* (one *PR5* family member) was rapidly induced by SCR96 challenge and increased over the time course (Figure 4). Remarkable induction by SCR96 treatment was found for three genes (*Solyc01g106605.1, Solyc04g007760.3*, and *Solyc09g014580.3*) that were not detected in the 0 hpt sample.

### 2.11. Genes Related to Transporter Activity

The GO analysis showed that transporter activity (GO:0005215) was one of the top enriched molecular function terms in all groups (Table 1). This term includes many descendants (child terms), such as ABC transporter (GO:0015424), one of the largest families of membrane proteins [28]. These transporters play important roles in diverse physiological processes, including pathogen responses [28]. The expression levels of 36 genes encoding transport proteins were found to be altered in the study (Appendix A; Appendix A). Of these 36 genes, 13 tended to be induced by SCR96 treatment, but most (64%) were suppressed, irrespective of time points. The upregulated genes were annotated with the ABC transporters (*Solyc01g068640.3, Solyc03g113040.4, Solyc03g113070.4, Solyc06g051730.4, Solyc06g070960.3,* and *Solyc07g008610.3*), amino acid transport and metabolism (*Solyc01g103030.3, Solyc09g011400.1*), glycerol-3-phosphate transporter 5 (*Solyc02g063020.3*), adenine/guanine permease (*Solyc03g111400.1, Solyc06g050250.1*), aquaporin TIP-type RB7-5A (*Solyc06g060760.3*), and organic solute transporter (*Solyc07g063520.3*) functions.

## 3. Discussion

### 3.1. Heterologous Expression of the Hydrophobic Effector Using a Newly Adapted Yeast System

For this study, we utilized an RNA Seq-based transcriptomic approach to investigate the gene regulation of tomato in response to the phytotoxic effector SCR96 from *P. cactorum*. To treat tomato leaves, we implemented the *P. pastoris* PichiaPink Strain 4 expression platform to express this hydrophobic small protein. Prior expression efforts in other systems resulted in low soluble yields of protein due to aggregation induced by a high percentage of cysteine residues [5,29]. While low yields of PcF with six SS bridged cysteines were also reported with the use of a bacterial expression system [30], we found that this newly adapted yeast system could yield recombinant protein in active form in a more cost-effective manner than when using either the mammalian cells or bacterial expression systems that we employed previously [5,31]. Although the mammalian cell protein expression system could yield similar amounts of purified protein in active form, the system was prohibitive in terms of cost [5]. In contrast, the recombinant protein produced by the bacterial expression system lacked bioactivity [31], although the yield was the highest. Using the yeast system described here, from 1 L of cell culture, about 140 μM of the purified SCR96 could be obtained in active form with relative ease and in a cost-effective manner. Hence, this method provides an alternative way to obtain recombinant proteins for hydrophobic small secreted effectors of oomycetes, such as SCRs.

### 3.2. SCR96 Induces Plant Defense Responses

In modern agriculture, development of resistant cultivars is the best approach to combatting plant diseases, including *Phytophthora* blight [32]. Breeding of resistant plants through gene engineering requires an in-depth understanding of the mechanisms of interaction between host plants and pathogens at the molecular level. It is well-known that *Phytophthora* pathogens secrete an array of apoplastic and cytoplasmic effectors to manipulate the physiological and defense networks of host plants [3]. The PcF/SCR effectors identified so far are derived from different *Phytophthora* species and believed to play important roles during *Phytophthora*–plant interactions [4,11,12,13]. This protein family has been regarded as one class of phytotoxic proteins since the discovery of the first PcF member based on phytotoxicity [11,13,30]. However, only a fraction of the genes from this family, including SCR96 and SCR82, have been assigned a biological function [4,5,16,33]. Silencing or knockout of such genes as *scr96* in *P. cactorum* and *scr82* in *P. capsici* led to decreased virulence and oxidative stress tolerance in the pathogens, indicating that these effectors are virulence factors [4,16]. However, previous work demonstrated that PcF/SCR effectors not only cause plant cell death, probably contributing to the pathogens’ virulence, but also trigger activation of the key defense-related phenylalanine ammonia lyase and other PR genes [11,14,16,30]. The present study aimed to reveal the global transcriptome profile of SCR96-induced response in plant. Our data revealed the transcriptional changes in numerous defense-related and metabolic genes in response to SCR96 elicitation. Specifically, we observed differential expression of several pattern recognition receptor and NLR genes post-SCR96 treatment. Such data thus blur the distinction between the responses caused by SCR96 belonging to PTI and ETI. This is quite similar to the many increasing examples, such as *P. sojae* XEG1 [34,35], indicating that boundaries between these phases are not distinct but rather blended, pointing to a PTI–ETI continuum [36,37]. Recently, the continuous crosstalk between PTI and ETI pathways in plants was reported [38,39], indicating that plant responses caused by SCR96 are worthy of further dissection.

Previously, PcF/SCR effectors were also regarded as oomycete Avr proteins, as they possess similar features as fungal Avr proteins [10,13,15,40]. However, no gene-for-gene model has been described so far [6,30]. Recently, the receptor of *P. infestans* SCR74 was mapped to a 43 kbp *G-LecRK* locus in wild potato [17], demonstrating a genetic method that could be used to untangle the mysterious nature of this family. This inspired us to check if *G-LecRK* genes were SCR96-responsive. It turned out that a number of *G-LecRK* genes were differentially expressed during SCR96 treatment, making it difficult to determine the response specificity of the gene locus to SCR96. Unlike SCR74, SCR96 does not contain the PcF domain [26] and, therefore, may have a different mechanism of action. The identification and characterization of the plant targets, receptors, and substrates of the effectors may help us better understand their biological functions and plant defenses during infection.

### 3.3. Signaling Triggered by SCR96 Can Reprogram the Expression of Genes Encoding Proteins Involved in Regulation of Plant Immunity

TFs form a repertoire of master regulators in the control of various processes relating to plant development and responses against external stimuli. It is well-known that TFs are involved in the regulatory interplay between plants and pathogens. Many TF families, including the AP2/ERF and MYB families, have been reported to be differentially expressed in plants as a reaction to pathogen infections, mediating their downstream target signaling pathways in plant defense [41,42]. In the current study, more than two thousand genes from the 76 TF families were identified as significantly participating in transcriptional regulation after SCR96 treatment. This discovery indicates that SCR96 triggers the activation of plant defense responses against pathogens. Similarly, the bacteria-, fungi-, and oomycete-derived proteins or PAMPs can also stimulate plant immunity, regulating a network of signaling pathways that fine-tune transcriptional activation of defense-related genes [36,43,44].

Strikingly, most of these TFs were upregulated in the three pairwise comparisons, as shown in H3 vs. H0, H6 vs. H0, and H12 vs. H0, revealing that transcriptional activation—but not repression—was significantly involved in the SCR96 response. Such TF genes include those of the AP2/ERF, MYB, and HB families. The AP2/ERF family is a multigene family of transcription factors unique to plants. They play a variety of roles throughout the plant life cycle, from being key regulators of developmental processes to forming part of the mechanisms used by plants to respond to various types of biotic and environmental stress [45,46]. For example, some members possess only a single DNA-binding domain, the expression of which is regulated by plant hormones, such as jasmonic acid, salicylic acid, and ethylene, as well as by pathogen challenge [47,48]. The MYB protein family is extraordinarily diverse in higher plants and known to be involved in the regulation of disease-resistance pathways [42]. The HB family controls many developmental pathways and physiological processes in eukaryotes. They are reported to be differentially regulated by drought stress in different crops [49]. However, little was known about their role in plant disease resistance until recently, when it was discovered that suppression of the HB gene HDTF1 enhances resistance to *Verticillium dahliae* and *Botrytis cinerea* in cotton [50]. Whether this family is involved in plant defense responses against *Phytophthora* pathogens needs further analysis. In sum, these results indicate that the complicated signaling of transcription factors could be initialized by SCR96 treatment and transduced to defense responses.

### 3.4. Importance of Early Interaction in Determining Compatibility

Early interaction events involving plants and pathogens often determine the final outcomes of the defense and counter-defense battles. The first layer of plant immunity—i.e., PTI triggered at the early interaction stage—can restrict the growth of the vast majority of pathogens [51]. Previous studies have mainly focused on transcriptomic responses during the early phase of plant responses to bacterial and fungal PAMPs or proteins [43,44,52]. In this study, the differential expression of plant genes related to defense mechanisms was investigated at 3, 6, and 12 h after SCR96 treatment to cover the biotrophic period of *Phytophthora* infections [2,53]. It was found that most of the genes categorized into different groups were significantly induced within the first 3 hpt. This demonstrated that the tomato plants had already mounted extensive reactions at this time point. Moreover, the tremendous differences in the responses to the phytotoxic protein between the three comparisons (H3 vs. H0, H6 vs. H0, H12 vs. H0) indicated that the early interaction was very likely to determine the subsequent defense cascades leading to either resistance or susceptibility.

## 4. Materials and Methods

### 4.1. Plant and Microbe Strains

*Solanum lycopersicum* cv. L402 (tomato) was cultivated at 25 °C under a 16 h/8 h light/dark rhythm in a greenhouse. Seedlings were maintained under the same conditions during SCR96 treatments.

*P. pastoris* PichiaPink Strain 4 (Thermo Fisher Scientific, Waltham, MA, USA) was maintained at 30 °C in YPD medium (1% (*w*/*v*) yeast extract, 2% (*w*/*v*) peptone, 2% (*w*/*v*) dextrose). *Escherichia coli* strain JM109 was grown at 37 °C in Luria-Bertani medium (1% (*w*/*v*) tryptone, 0.5% (*w*/*v*) yeast extract, 0.5% (*w*/*v*) NaCl).

### 4.2. Nucleic Acid Manipulation

Total RNA was extracted from tomato leaves using RNAiso Plus reagent (TaKaRa, Kusatsu, Shiga, Japan) following the manufacturer’s instructions. RNA concentration and purity were measured using a NanoDrop 2000 (Thermo Fisher Scientific, Waltham, MA, USA). RNA integrity was assessed using the RNA Nano 6000 Assay Kit in an Agilent Bioanalyzer 2100 system (Agilent Technologies, Santa Clara, CA, USA). The RNA samples were treated with DNase I (TaKaRa, Kusatsu, Shiga, Japan) and reverse transcribed into first-strand cDNAs using M-MLV reverse transcriptase (RNase H Minus) and oligo (dT)_18_ primers (TaKaRa, Kusatsu, Shiga, Japan).

### 4.3. Plasmid Construction

To facilitate the expression of *scr96* (GenBank no. KT215393) in *P. pastoris* cells, the gene codon, excluding its signal peptide, was optimized using the OptimumGene algorithm (GenScript, Nanjing, Jiangsu Province, China). The codon-optimized gene fragment (Appendix A) was synthesized by GenScript and cloned into a modified pPink-HC (Thermo Fisher Scientific, Waltham, MA, USA) containing a *Saccharomyces cerevisiae* α-mating factor secretion signal and C-terminal His_9_ tag using *Nco*I and *Kpn*I restriction sites. The resultant plasmid was verified by endonuclease digestion and DNA sequencing.

### 4.4. Yeast Expression and Protein Purification

The expression plasmid was introduced into the PichiaPink Strain 4 cells following a previously described protocol [54] with minor modifications. Briefly, the electroporating pulse was applied with a 2 mm cuvette at 2 kV, 25 μF, and 200 Ω using GenePulser Xcell (Bio-Rad, Hercules, CA, USA). After pulsing, the cells were immediately re-suspended in 1 mL of ice-cold 1 M sorbitol in the cuvette and incubated for 10 min on ice. Then, the uncapped cuvette was kept at room temperature for 6 h. Cells in a 200 μL aliquot were spread on Adenine dropout medium. After 3–4 d, the white colonies were analyzed using colony PCR to determine if the gene of interest had integrated into the yeast genome.

The positive yeast clones were inoculated in 5 mL of buffered sorbitol-complex media (BMSY) (1% yeast extract, 2% peptone, 100 mM potassium phosphate buffer (pH 6.0), 1.34% yeast nitrogen base (YNB) with ammonium sulfate without amino acids, 0.4% biotin, 1% sorbitol, and 0.1% trace element solution (2.2% ZnSO_4_•7H_2_O, 1.1% H_3_BO_3_, 0.5% MnCl_2_•4H_2_O, 0.5% FeSO_4_•7H_2_O, 0.16% CoCl_2_•5H_2_O, 0.16% CuSO_4_•5H_2_O, 0.11% (NH_4_)_6_Mo_7_O_24_•4H_2_O, and 5% Na_4_EDTA)) and incubated at 30 °C and 240 rpm for 24 h. The culture was transferred into 1 L BMSY in a 2 L flask, covered with three-layer sterile gauze, and grown at 30 °C and 240 rpm. After 48 h growth, the culture was harvested in sterile centrifuge bottles at 10,000× *g* for 15 min at room temperature. Yeast cells were re-suspended in 330 mL sterile water, then supplemented with 160 mL protein expression media (PEM) (100 mM potassium phosphate buffer (pH 6.0), 1.34% YNB, 0.4% biotin, 1% sorbitol, 0.1% trace element solution, 2% methanol). Ten milliliters of 100% methanol was added to the mixture last and swirled immediately. The cells were incubated at 30 °C and 240 rpm for 24, 48, 72, or 96 h. Every 24 h, PEM-Daily Additions Mixture (0.2% sorbitol, 0.02% trace element solution) was added to the culture. Every 12 h, methanol (making up 0.5% of the total volume) was replenished to compensate its loss. At 24 h intervals, 1 mL of the culture was taken to check for protein induction.

When the protein yield peaked, the induction was concluded with protein purification using centrifugation at 11,000× *g* for 15 min. The recombinant protein fused with the C-terminal His_9_ tag was purified from the culture supernatant with Ni–NTA affinity resin as previously described [5].

### 4.5. Transcriptome Studies

Tomato seedlings were treated by infiltrating an aqueous SCR96 solution (500 nM) into leaves. At 0, 3, 6, and 12 hpt, leaves were harvested in three biological replicates. RNA isolation from tomato leaves and a quality check were performed as described above.

A total amount of 1 μg RNA per sample was used as input material for the RNA sample preparations. The 12 sequencing libraries were generated using an NEBNext Ultra II Directional RNA Library Prep Kit for Illumina (NEB, Ipswich, MA, USA) following manufacturer’s protocol. All samples were sequenced with the Illumina NovaSeq 6000 platform with the pair-end mode at Genepioneer Biotechnologies (Nanjing, China). The raw reads generated from the experiments were deposited in the NCBI Sequence Read Archive (PRJNA830601).

Clean reads were obtained by removing low-quality reads containing adapter or polyN from raw data through in-house Perl scripts. All clean reads from all biological experiments were mapped to the tomato ITAG4.0 genome [55] with HISAT2 (http://ccb.jhu.edu/software/hisat2/manual.shtml; accessed on 14 December 2019). Gene expression levels were estimated in fragments per kilobase of transcript per million fragments mapped (FPKM). Transcript abundance and differential gene expression were calculated with the DESeq R package (v1.10.1) as previously described [56]. The *p* values were adjusted using the Benjamini–Hochberg approach for controlling the false discovery rate. Genes with an adjusted *p* < 0.01 and absolute value for the log2(fold change) >1 were assigned as DEGs. The multiple comparisons of DEG sets were performed and visualized using the UpSetR package [57] in R. The expression profile analysis and hierarchical clustering of unique DEGs were conducted using the function heatmap.2 in the gplots [58] package in R.

Gene functions were annotated based on the National Center for Biotechnology Information (NCBI) NR, GO, COG, KOG, KEGG, Swiss-Prot, and Pfam databases with a 10^−5^ cut-off *E* value, as previously described [59].

GO and KEGG pathway enrichment analyses were conducted as previously described [59]. To identify transcripts with similar accumulation profiles, a clustering analysis using the Clust tool [22] was performed. The RPKM values were log-transformed and used to generate the heatmaps with TBtools [60] with the hierarchical clustering method.

### 4.6. Validation Analysis Using qRT-PCR

Tomato leaf treatment and RNA and cDNA preparation were performed using the same conditions as described for the transcriptome analysis. qRT-PCR was performed with cDNA samples using ChamQ SYBR qPCR Master Mix (High ROX Premixed) (Vazyme Biotech, Nanjing, Jiangsu Province, China) and gene-specific primers (Appendix A) in a CFX Connect Real-Time PCR detection system (Bio-Rad, Hercules, CA, USA). Briefly, the 20 μL reaction volume consisted of 10 μL qPCR master mix containing SYBR Green, 1 μL cDNA, and 0.2 μM of each primer and was incubated with the following program: 95 °C for 30 s followed by 40 cycles of 95 °C for 10 s, 60 °C for 30 s, and a final melting curve analysis (95 °C for 15 s, 60 °C for 60 s, 95 °C for 15 s). Expression levels were calculated using the threshold cycle (Ct) values for each gene with the Relative Expression Software Tool (REST) [61]. PCR efficiency for each reaction was obtained from the slope of the standard curves, where efficiency = 10^−1/slope^ [62]. The tomato *actin* gene [63] was used as the reference gene for normalizing Ct values. The 0 hpt cDNA sample served as a calibrator and was set at the value 1. Three independent experiments were performed.

## 5. Conclusions

This study demonstrated that changes in the expression of tomato defense-related genes occurred over the time course of a treatment with the small, phytotoxic apoplastic effector SCR96 from the oomycete phytopathogen *P. cactorum*. Upon treatment with SCR96, elicitor perception and a series of signal transductions and defense responses were activated. The most extensive changes by far occurred at the early interaction stage, probably indicating a sensitive and comprehensive response from the tomato transcriptome to pathogen invasion. We also observed changes in genes involved in transport activity that may help establish the development of a defense barrier. In summary, our study revealed many candidate genes involved in this response that may be useful for engineering crop resistance against destructive *Phytophthora* pathogens.

## Figures and Tables

**Figure 1 plants-12-00883-f001:**
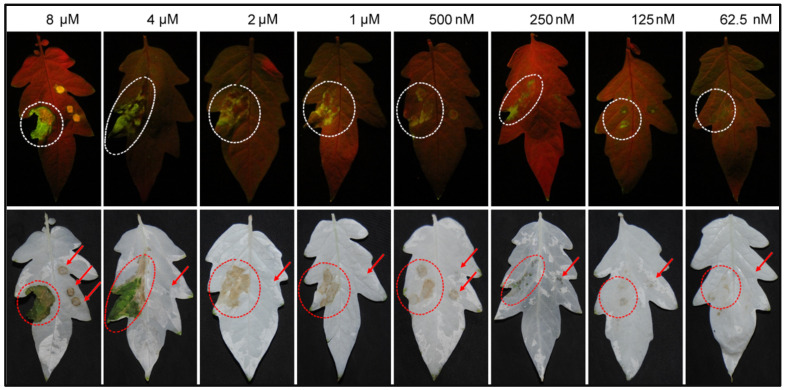
**The recombinant SCR96 protein caused necrosis in tomato leaves.** The leaves of tomato cultivar L402 were infiltrated with the recombinant protein solution and photographed at 12 h post-infiltration under UV light (upper panel) and under natural light after ethanol destaining (lower panel). The dotted circles on the left sides of the leaves indicate the lesions caused by SCR96. In contrast, concurrent infiltration with PBS on the right sides did not produce a host response. Dot traces (indicated by arrows) on the right sides of the leaves were occasionally visible (i.e., 8 μM and 500 nM panels), which were caused by the syringe orifices without needles during the infiltration process.

**Figure 2 plants-12-00883-f002:**
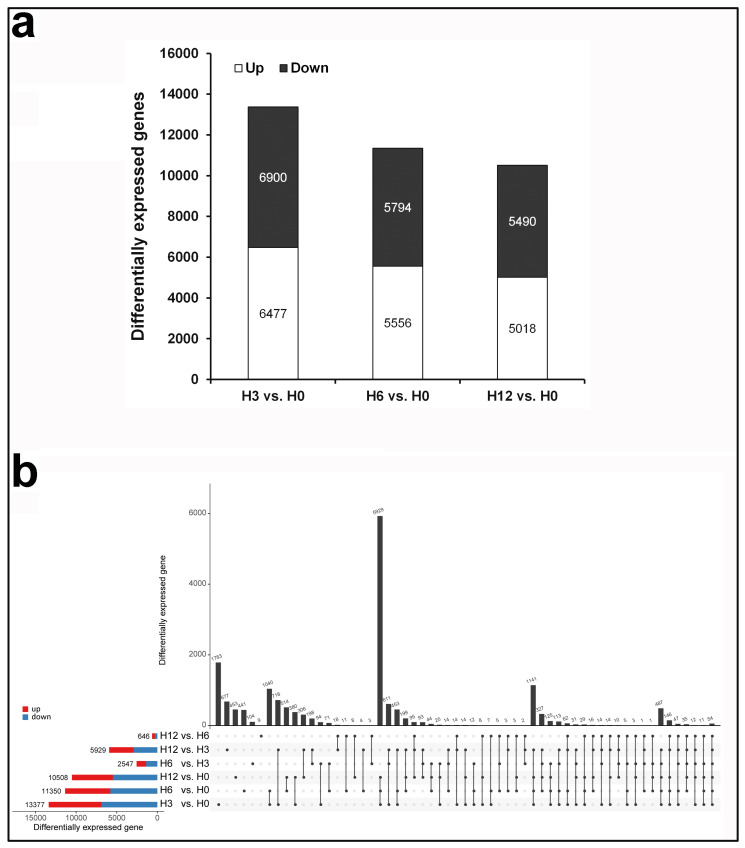
**Multiple comparisons of DEG datasets.** (**a**) Number of DEGs per dataset. Bars represent up- (white) and downregulated (black) genes in the three pairwise comparisons. (**b**) The UpSet R plot showing numbers of DEGs in each set and each comparison.

**Figure 3 plants-12-00883-f003:**
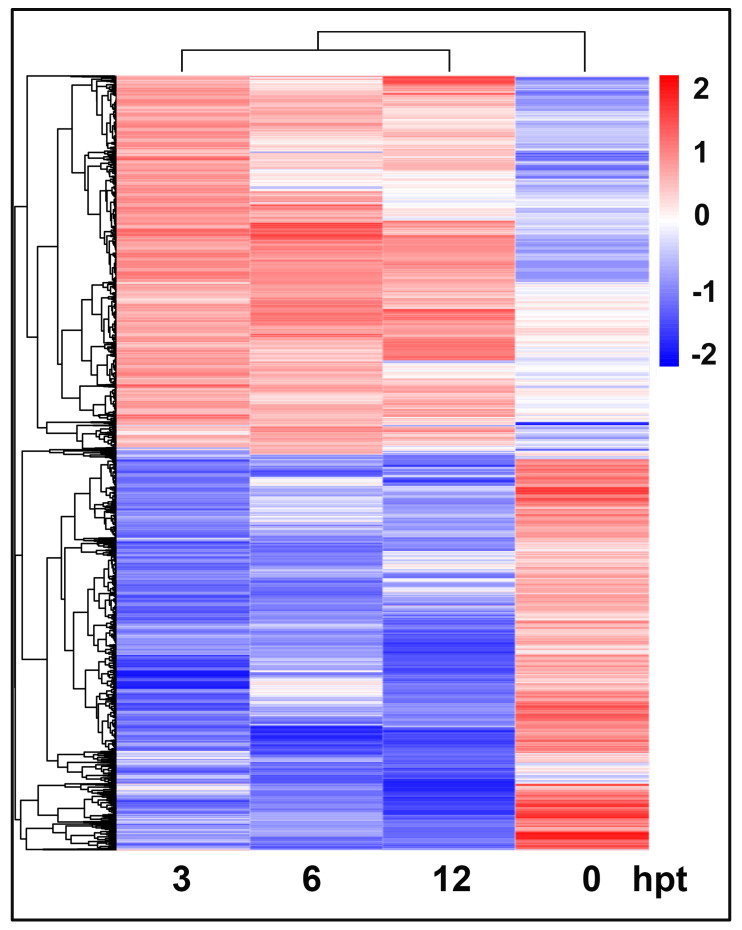
Heat map and dendrograms of hierarchical clustering showing the expression patterns of all DEGs in response to SCR96 treatment at four time-points (0, 3, 6, 12 hpt). The colors represent the upregulation (red) and downregulation (blue) of statistically significant DEGs (log2|fold change| > 1, *p* < 0.01) with different magnitudes (brightness) for the log2 fold change values. The dendrogram on top shows the clustering of time points. The dendrogram on left side shows the clustering of DEGs.

**Figure 4 plants-12-00883-f004:**
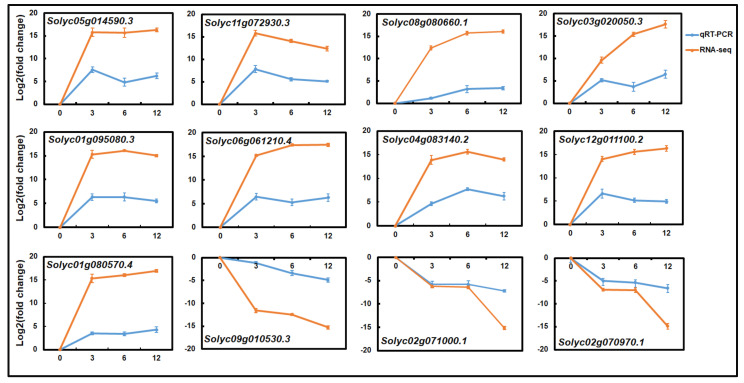
**The changes in the expression levels of 12 genes regulated by SCR96 over the time course were confirmed with qRT-PCR.** Expression level values of each gene were normalized to the expression level values of the housekeeping *actin* gene and transformed using log2 as for the RNA-Seq fragments per kilobase of transcript per million fragments mapped (FPKM) values. The assays were repeated three times and showed similar expression patterns. Bars indicate standard errors.

**Figure 5 plants-12-00883-f005:**
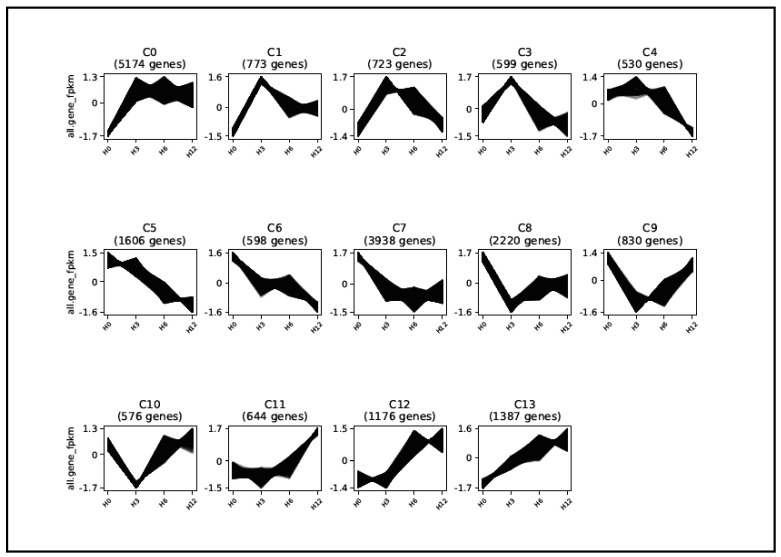
**SCR96-responsive genes could be clustered based on temporal expression patterns.** SCR96-responsive genes were clustered using Clust according to their expression kinetics. Fourteen major clusters were obtained. The temporal expression profiles of the genes in each cluster are represented by line graphs. The y-axis of each graph shows the normalized expression value of each gene in FPKM, and the x-axis represents the corresponding time points (H0, H3, H6, and H12 mean 0, 3, 6, and 12 h) after SCR96 treatment. The numbers in parentheses are the numbers of genes within each cluster.

**Figure 6 plants-12-00883-f006:**
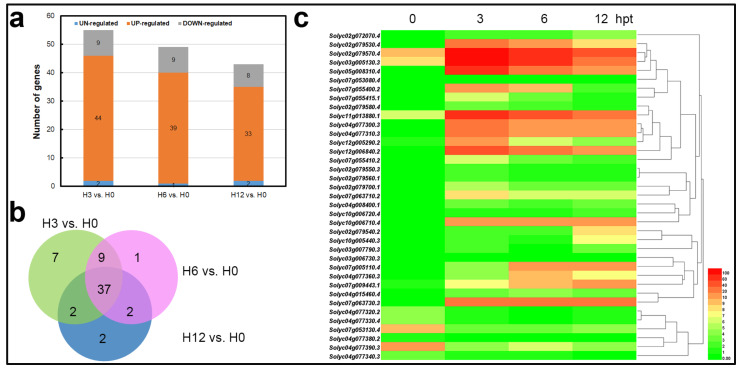
**Expression profiles of *G-LecRK* genes responding to SCR96 treatment.** (**a**) The bar graph represents the numbers of *G-LecRK* genes that were up-, down-, or unregulated by SCR96 in the three comparison sets (H3 vs. H0, H6 vs. H0, H12 vs. H0). (**b**) Venn diagram showing the overlap between the three comparison sets. (**c**) Heat map and dendrograms of hierarchical clustering showing the expression patterns of all 37 *G-LecRK* DEGs at 0, 3, 6, and 12 hpt in response to SCR96.

**Figure 7 plants-12-00883-f007:**
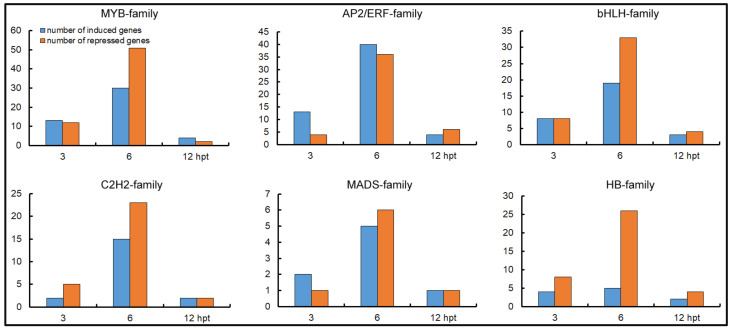
Expression patterns of six important TF families regulated by SCR96. Genes with mRNA abundance induced or repressed by more than twofold were grouped into families.

**Table 1 plants-12-00883-t001:** The overrepresented terms from GO categories identified in the DEGs in the three comparisons (H3 vs. H0, H6 vs. H0, H12 vs. H0).

Category	GO Term	Functional Description	False Discovery Rate (FDR) ^a^		
			H3 vs. H0	H6 vs. H0	H12 vs. H0
*Biological processes*					
	GO:0006952	Defense response	0.0005	2.84 × 10^−5^	0.0008
	GO:0051179	Localization	0.017	0.010	0.007
	GO:0009607	Response to biotic stimulus	0.061	0.044	0.027
	GO:0060154	Cellular process regulating host cell cycle in response to virus	0.036	0.031	―
	GO:0002376	Immune system process	0.036	0.029	0.029
	GO:0055114	Oxidation–reduction process	0.078	0.043	0.022
	GO:0008152	Metabolic process	0.026	0.042	0.026
*Cellular component*					
	GO:0016020	Membrane	0.0002	7.71 × 10^−5^	0.0003
	GO:0005576	Extracellular region	0.099	0.099	0.088
	GO:0043226	Organelle	0.035	0.031	―
	GO:0043234	Protein-containing complex	0.047	0.034	0.091
	GO:0030054	Cell junction	0.019	0.017	0.015
*Molecular function*					
	GO:0000989	Transcription factor binding	0.063	0.028	0.066
	GO:0005102	Signaling receptor binding	0.059	0.053	0.049
	GO:0030674	Protein binding and bridging	0.044	―	0.058
	GO:0001664	G-protein-coupled receptor binding	0.005	0.034	0.014
	GO:0003676	Nucleic acid binding	0.013	0.077	0.0091
	GO:0003824	Catalytic activity	0.027	0.07	0.02
	GO:0005215	Transporter activity	0.039	0.043	0.032

^a^ The analysis of GO term enrichment was performed for DEGs at each time point in comparison to genes in the whole genome as a background using hypergeometric tests with an FDR cutoff of 0.05.

## Data Availability

The sequencing raw data are available from the NCBI Sequence Read Archive (SRA) under accession number PRJNA830601.

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
