# Peer review of "Time-Course Transcriptome Profiling Reveals Differential Resistance Responses of Tomato to a Phytotoxic Effector of the Pathogenic Oomycete Phytophthora cactorum"

_plants, 2023, doi:10.3390/plants12040883_

Round 1

Reviewer 1 Report

The manuscript is dealing with transcriptome profiling of tomato seedlings exposed to treatment with heterologously expressed SCR96 protein which was previously identified by the same research group as a virulence factor of the oomycete Phytophthora cactorum. The paper interesting and novel findings and the results are convincingly presented.
I am not concerned about the scientific content of the paper. However, the text doesn't read well and it needs a thorough language editing, styling and and partial re-organizing. The Introduction is too long and includes a large amount of side-information. The authors should consider to move some parts of the Introduction to the Discussion. Further, the should take more care about clear separation of contents belonging to the Methods and Results sections.
Finally, it is advisable that the authors avail themselves on a professional manuscript editing service.

Author Response

Reviewer 1:

The manuscript is dealing with transcriptome profiling of tomato seedlings exposed to treatment with heterologously expressed SCR96 protein which was previously identified by the same research group as a virulence factor of the oomycete Phytophthora cactorum. The paper interesting and novel findings and the results are convincingly presented.

I am not concerned about the scientific content of the paper. However, the text doesn't read well and it needs a thorough language editing, styling and and partial re-organizing. The Introduction is too long and includes a large amount of side-information. The authors should consider to move some parts of the Introduction to the Discussion. Further, the should take more care about clear separation of contents belonging to the Methods and Results sections.

Finally, it is advisable that the authors avail themselves on a professional manuscript editing service.

A: Thanks for your detailed comments and suggestions.

One of our co-authors, Dr. Shiv D. Kale, is an English-native speaker (USA scientist), who has very strong academic background in the field. He has critically edited the texts of our manuscript thoroughly. Now we made extensive revisions to the manuscript by considering your comments and those from other reviewers.

We deleted one paragraph and rephrased the sentences in the introduction part to eliminate the side-information.

We carefully re-organized the texts and separated the MM part from Result part as much as we could. We also think some summary introduction is necessary because we need to remind readers there. 

We also made necessary changes to the manuscript by considering your suggestions listed in the sheet.

Reviewer 2 Report

The manuscript is devoted to the actual topic of the pathogen-host interaction system on the example of one of the economically important crop; tomato.

The article is well structured, illustrated and presented.

Сomments:

1 All abbreviations must be spelled out at the first mention in the test. Authors need to check the text and make the appropriate corrections.

2. Change the beginning of the sentences on L 307-310 “Of them, ……” and L 369 “Of these…”, because it is not entirely successful.

3. L 380-392 - the test is more in line with the “Results” section.

4 In the section “Materials and Methods”:

L 481-2 - it is necessary to specify where the plant’s material was grown: in the open field conditions or in a greenhouse

L486- it is necessary to add the reference, where the corresponding laboratory method is described

Author Response

Сomments:

1. All abbreviations must be spelled out at the first mention in the test. Authors need to check the text and make the appropriate corrections.

A: Thanks for your suggestions. We made the revisions as suggested.

2. Change the beginning of the sentences on L 307-310 “Of them, ……” and L 369 “Of these…”, because it is not entirely successful.

A: Thanks for your suggestions. We made the revisions to such sentences as suggested.

3. L 380-392 - the test is more in line with the “Results” section.

A: Thanks for your comment. To state that we adapted a yeast expression platform that is much better than other systems, we need to introduce the background information, and compare them from different aspects So that readers can follow the story. Please read and try to understand the texts.

4. In the section “Materials and Methods”:

L 481-2 - it is necessary to specify where the plant’s material was grown: in the open field conditions or in a greenhouse

A: Thanks for your suggestions. We made the revision as suggested.

L486- it is necessary to add the reference, where the corresponding laboratory method is described

A: Thanks for your suggestions. We revised the sentence.

We also made necessary changes to the manuscript by considering your suggestions listed in the sheet.